# PLANAR HOMEOMORPHIC EMBEDDINGS OF DECISION TREE

## ABSTRACT

Decision trees and their ensemble variants are widely celebrated for their accuracy, interpretability, and effectiveness on tabular data. Despite their intuitive structure, understanding the global geometric organization of the feature space partitions induced by these models remains challenging, particularly in high-dimensional settings. Traditional visualization techniques, such as node-link diagrams, fail to capture the topological relationships between decision regions, while standard dimensionality reduction methods prioritize data distribution over structural fidelity, often distorting adjacency and connectivity. To address this limitation, we propose a novel framework for embedding decision-tree-induced partitions into two-dimensional space while explicitly preserving adjacency relations among leaf regions. Our approach models the decision tree as a polyhedral complex and constructs a piecewise-linear (PL) embedding that maintains the combinatorial topology of the original high-dimensional partitioning. This adjacency-preserving visualization enables a more faithful interpretation of model behavior, revealing insights into decision boundary structure and data distribution. Our theoretical and experimental results demonstrate the feasibility of the proposed method and its ability to preserve the topological characteristics of the data.

## 1 INTRODUCTION

Decision trees and their ensembles (e.g., random forests, gradient boosting machines) are among the most widely used models in machine learning due to their balance of accuracy, interpretability, and scalability Breiman (2001); Friedman (2001), especially on tabular data. Unlike black-box models such as deep neural networks, decision trees partition the feature space into axis-aligned regions, each corresponding to a leaf node with an associated prediction. This partition-based structure has made them especially valuable in domains such as healthcare, finance, and scientific discovery, where understanding the decision process is as critical as predictive accuracy Caruana et al. (2015); Rudin (2019).

Yet despite their conceptual simplicity, visualizing and interpreting decision trees in high-dimensional settings remains a challenging task. Each tree induces a subdivision of the feature space into adjacent polyhedral regions, forming a complex geometric structure. While individual decision paths are easily interpretable, the global organization of the partitions—how different regions interact, where decision boundaries meet, and how predictions vary across boundaries—remains opaque when the feature dimension exceeds three Hastie et al. (2009). This makes it difficult to answer questions such as: Which regions are adjacent in the input space? Where are predictions most sensitive to perturbations? How is the data distributed in space?

Existing visualization techniques only partially address these challenges. Classical approaches such as node-link diagrams Louppe (2014) capture the branching structure of the tree but discard information about the geometry of the induced partitions. Low-dimensional projections (e.g., 2D feature slices or partial dependence plots Friedman (2001)) provide local snapshots of decision boundaries but fail to convey the global adjacency relations between partitions. Dimensionality reduction methods such as PCA Abdi & Williams (2010), t-SNE Maaten & Hinton (2008), and UMAP McInnes et al. (2018) have been successfully applied to visualize high-dimensional data, yet they are designed to preserve variance, neighborhood density, or manifold structure, rather than the combinatorial and topological properties of model-induced partitions. Consequently, when applied to decision trees, these methods often distort adjacency, merge distinct regions, or artificially separate continuous regions, limiting their utility for faithful visualization and interpretation.

In this work, we address this gap by proposing a novel framework for embedding decision-tree-induced partitions from high-dimensional spaces into two-dimensional domains while explicitly preserving adjacency rela-

tions among regions. Our method treats the decision tree as a geometric complex: each leaf corresponds to a cell, and adjacency is defined by shared boundaries in the original space. We then construct a piecewise-linear (PL) embedding that maps this complex into the plane, ensuring that adjacent cells in the high-dimensional space remain adjacent in the embedding. Unlike conventional dimensionality reduction techniques, our approach prioritizes structural fidelity over point-level similarity, yielding embeddings that are topologically faithful to the original model. This adjacency-preserving embedding enables new possibilities for model interpretability and analysis. For practitioners, it provides an intuitive visualization that reveals how regions of the feature space are organized, how decision boundaries intersect, and where the model may be overly sensitive to perturbations. For researchers, it bridges combinatorial representations of decision trees with geometric embeddings of high-dimensional complexes, connecting machine learning with tools from computational topology and piecewise-linear geometry Edelsbrunner & Harer (2010).

Our contributions are threefold:

- Adjacency-preserving embedding framework. We introduce the first principled approach for mapping decision-tree partitions into two dimensions while preserving adjacency relations between regions.

- Constructive algorithm. We design efficient approximate embedding algorithms that guarantee structural fidelity of the decision tree complex.

- Experimental demonstration. We demonstrate the utility of our embeddings in visualization and decision boundary analysis, showing that they provide richer insights and topological characteristics preservation than tree visualization methods.

## 2 RELATED WORK

Decision trees: Decision trees are a fundamental family of supervised-learning algorithms for both classification Breiman (2001); Kotsiantis (2013); Quinlan (1986) and regression Bertsimas et al. (2017); Loh (2011). Structurally, a tree comprises a single root node, a set of internal (test) nodes, and a collection of terminal (leaf) nodes. Starting from the root, the instance space is recursively partitioned by means of axis-parallel, univariate splits of the form "$x^j \leq t$," each chosen to maximize the homogeneity of the resulting child subsets with respect to the target variable. Homogeneity is most commonly quantified through impurity-reduction criteria Raileanu & Stoffel (2004) such as information gain Kent (1983), Gini index Strobl et al. (2007), or variance reduction Suen et al. (2005), although any monotonic measure of node impurity may be employed. The recursive partitioning process induces a hierarchical tessellation of the feature space into axis-aligned hyper-rectangles or semi-infinite hyper-rectangles, every leaf being assigned a single class label (classification) or a constant prediction (regression).

Homeomorphism: $X$ and $Y$ are topological spaces. A mapping $h : X \rightarrow Y$ is called a homeomorphism Archdeacon (1996) if $h$ satisfies the following conditions: 1. It is a 1-1 mapping (in this case, injective); 2. It is surjective; 3. It is continuous, that is, it preserves the proximity properties of each point. 4. The inverse mapping is also continuous, and the topological properties of the homeomorphism are preserved. This paper aims to map the decision partitions of a high-dimensional decision tree to two dimensions while preserving its adjacency structure.

## 3 PRELIMINARIES

To facilitate the subsequent description, we first define the partitions formed by the decision tree and define the decision tree accordingly.

**Definition 3.1** (Polyhedral Complex). A *polyhedral complex* $\mathcal{C}$ in $\mathbb{R}^d$ is a finite collection of convex polyhedra (called *cells*) such that:

1. If $P \in \mathcal{C}$, then every face of $P$ is also in $\mathcal{C}$.

2. The intersection of any two polyhedra $P, Q \in \mathcal{C}$ is a face of both $P$ and $Q$.

The *underlying space* of $\mathcal{C}$, denoted $|\mathcal{C}|$, is the union of all its cells.

**Definition 3.2** (Decision-Tree Complex). Let $\mathcal{T}$ be a binary decision tree of depth $\Delta$ on $X \subset \mathbb{R}^d$. Each internal node $\nu$ is labelled by an affine function $f_\nu(x) = w_\nu^\top x + b_\nu$ with $w_\nu \in \{\pm e_j\}$ in case (A) or arbitrary $w_\nu \in \mathbb{R}^d$ in case (B). The leaf regions $\{R_i\}$ are defined as

$$R_i = \bigcap_{\nu \in \mathrm{path}(i)} H_\nu, \quad H_\nu = \begin{cases} \{x : f_\nu(x) \leq 0\} & \text{if we turn left at } \nu, \\ \{x : f_\nu(x) > 0\} & \text{if we turn right at } \nu. \end{cases}$$

The *decision-tree complex* $\mathcal{C}_\mathcal{T}$ is the polyhedral complex whose $d$-cells are the closures $\overline{R_i}$ and whose $k$-cells are the $k$-dimensional faces of these polyhedra.

After defining the decision tree, we can define the partition it constructs as a graph. The definition of the graph is as follows:

**Definition 3.3** (Combinatorial Dual Graph). Let $G(\mathcal{T})$ be the graph whose vertices correspond to the $d$-cells (leaf regions) of $\mathcal{C}_\mathcal{T}$ and whose edges connect two vertices if the corresponding $d$-cells share a common $(d-1)$-face.

Let $\mathcal{T}$ be a decision tree trained on a bounded domain $X \subseteq \mathbb{R}^d$. The tree induces a finite cell complex $\mathcal{C}_\mathcal{T}$ whose $d$-cells are the leaf regions $\{R_i\}_{i=1}^L$ and whose $(d-1)$-cells are pieces of the split hyperplanes. We study when $\mathcal{C}_\mathcal{T}$ can be embedded into $\mathbb{R}^2$ *homeomorphically*, i.e., via an injective continuous map $h\colon |\mathcal{C}_\mathcal{T}| \hookrightarrow \mathbb{R}^2$ that preserves the incidence structure of the complex.

**Definition 3.4** (Planar Embedding of a Complex). A polyhedral complex $\mathcal{C}$ is said to be *PL embeddable* in $\mathbb{R}^2$ if there exists a piecewise-linear homeomorphism $h : |\mathcal{C}| \to \mathbb{R}^2$ that maps each cell of $\mathcal{C}$ to a polyhedral cell in $\mathbb{R}^2$ while preserving the incidence relations.

This work establishes fundamental limitations and possibilities for such embeddings, with implications for the visualization of decision boundaries and the topological complexity of learned models.

We consider the following two cases respectively:

(A) Axis-aligned boundary pieces only.

(B) Arbitrary piecewise–linear (PL) boundaries allowed in the plane.

# 4 AXIS-ALIGNED BOUNDARIES

We initially expected that any high-dimensional, axis-parallel decision-tree partition could be faithfully embedded into the plane while preserving both its combinatorial structure and its axis-aligned geometry—an outcome that would greatly facilitate visual interpretation and downstream analysis. Contrary to this expectation, the following theorem demonstrates that once the ambient dimension exceeds three, such an embedding is no longer universally attainable: there exist axis-parallel partitions that cannot be mapped into two dimensions without sacrificing the axis-parallel property.

**Theorem 4.1** (Obstruction for Axis-Aligned Embeddings). There exists a decision tree $\mathcal{T}$ of depth 3 such that $G(\mathcal{T})$ contains a $K_4$-minor. Consequently, $\mathcal{C}_\mathcal{T}$ cannot be embedded into $\mathbb{R}^2$ with only axis-aligned segments as boundaries.

Following Theorem 4.1, we have the following corollary, which is obvious because a decision tree constructed in a space of any dimension that is partitioned only once or twice can be mapped into a two-dimensional space without changing the topological properties of its partition.

**Corollary 4.2.** If depth $\Delta \leq 2$, then $G(\mathcal{T})$ is outer-planar and $\mathcal{C}_\mathcal{T}$ admits an axis-aligned planar embedding.

**Theorem 4.3** (Characterization for Axis-Aligned Embeddings). For a decision tree $\mathcal{T}$ with axis-aligned splits, the following are equivalent:

1. $\mathcal{C}_\mathcal{T}$ admits an axis-aligned planar embedding.

2. The depth $\Delta \leq 2$.

Theorem 4.3 shows that we can only construct axis-parallel partitions in two-dimensional space without changing the topology for decision trees with a depth of 2 or less. For deeper decision trees, there is no method to construct axis-parallel partitions in two-dimensional space without changing the topology. Can we use more relaxed conditions, such as not requiring the mapped partitions in two-dimensional space to be axis-parallel, to obtain a mapping that preserves the topology between partitions? We will discuss this in the next section.

## 5 ARBITRARY PL BOUNDARIES

The negative result from last section establishes a fundamental limitation: axis-aligned embeddings of decision tree complexes are only possible for shallow trees of depth at most 2. However, this restriction stems from imposing the additional constraint that boundaries in the embedding must remain axis-aligned. By relaxing this requirement and allowing arbitrary piecewise-linear (PL) boundaries, we can overcome this limitation and construct faithful planar embeddings for decision trees of arbitrary depth and complexity.

In this section, we will present an algorithm that can map decision trees in high-dimensional space to two-dimensional space without changing the topological properties.

### 5.1 SUBDIVISION-BASED EMBEDDING ALGORITHM

Our method for constructing a PL planar embedding of an arbitrary decision tree complex $\mathcal{C}_{\mathcal{T}}$ is based on a two-step process:

- Planarizing the 1-skeleton through strategic subdivision.
- Embedding the higher-dimensional cells in a compatible manner.

The approach constructs a planar embedding through a systematic process that transforms the high-dimensional decision tree complex into a topologically equivalent two-dimensional representation. The core insight is that while the original complex $\mathcal{C}_{\mathcal{T}}$ may be non-planar, we can always obtain a planar subdivision $\mathcal{C}'_{\mathcal{T}}$ that preserves the essential combinatorial structure through strategic edge subdivisions. This approach relies on two fundamental results:

**Definition 5.1** (Edge Subdivision Operator)**.** Let $G = (V, E)$ be a graph. The *subdivision* of an edge $e = (u, v) \in E$ yields a new graph $G' = (V \cup \{w\}, (E \setminus \{e\}) \cup \{(u, w), (w, v)\})$. A *k-subdivision* of $G$ is a graph obtained by applying a sequence of $k$ edge subdivisions to $G$.

**Lemma 5.1** (Planarization via Subdivision)**.** For any finite graph $G = (V, E)$, there exists an integer $K \leq |E|(|E| - 1)/2$ and a $K$-subdivision $\tilde{G}$ of $G$ that is planar.

Edge subdivision provides a mechanism to resolve graph crossings without altering the fundamental topology. Each subdivision effectively "breaks" an edge at crossing points, allowing the graph to be drawn without intersections while preserving connectivity.

Lemma 5.1 guarantees that any graph, regardless of its inherent non-planarity, can be transformed into a planar graph through a finite sequence of edge subdivisions. The upper bound $K \leq \binom{|E|}{2}$ establishes that the process is computationally tractable.

**Lemma 5.2** (Straight-Line Embedding)**.** Let $\tilde{G} = (V, E)$ be a planar graph obtained from the 1-skeleton of $\mathcal{C}_{\mathcal{T}}$ by subdivision. Then there exists an embedding $\phi : V \to \mathbb{R}^2$ such that:

1. For every edge $(u, v) \in E$, the line segment $[\phi(u), \phi(v)]$ is a straight line

2. For any two distinct edges $(u_1, v_1), (u_2, v_2) \in E$, the segments $[\phi(u_1), \phi(v_1)]$ and $[\phi(u_2), \phi(v_2)]$ intersect only if they share a common vertex

Theorem 5.2 ensures that our planarized graph can be embedded using only straight line segments. This property is crucial for constructing a piecewise-linear embedding that maintains geometric simplicity while preserving topological relationships. he complete embedding procedure is formalized in Algorithm 1[1].

---

[1] The code is avaliable in `https://anonymous.4open.science/r/PLPE`.

---

**Algorithm 1** PL Planar Embedding of Decision Tree Complex

---

1: **Input:** Decision tree $\mathcal{T}$, complex $\mathcal{C}_{\mathcal{T}}$
2: **Output:** Embedded subdivision $\mathcal{C}'_{\mathcal{T}} \subset \mathbb{R}^2$
3:
4: STEP 1: PLANARIZE THE 1-SKELETON
5: $G_0 \leftarrow$ 1-skeleton of $\mathcal{C}_{\mathcal{T}}$
6: $\tilde{G} \leftarrow \text{PlanarSubdivision}(G_0)$          ▷ Apply Lemma 5.1
7: $\phi \leftarrow \text{StraightLineEmbedding}(\tilde{G})$          ▷ Apply Lemma 5.2
8:
9: STEP 2: EMBED 2-CELLS
10: **for** each 2-cell $f$ in $\mathcal{C}_{\mathcal{T}}$ **do**
11:      $\partial f \leftarrow$ boundary cycle of $f$ (a closed walk in $\tilde{G}$)
12:      $P_f \leftarrow$ polygon in $\mathbb{R}^2$ formed by $\phi(\partial f)$
13:      SUBDIVIDE $f$ into triangles $T_1, \ldots, T_m$ such that $\partial T_i \subset \partial P_f$ or is a new interior edge
14:      MAP each new vertex $v$ in the interior of $f$ to a point in the interior of $P_f$ via barycentric coordinates relative to $T_i$
15: **end for**
16:
17: STEP 3: PROJECT HIGHER-DIMENSIONAL CELLS
18: **for** each cell $c$ of dimension $\geq 3$ **do**
19:      SUBDIVIDE $c$ into simplices $\{\sigma_i\}$
20:      PROJECT each $\sigma_i$ onto the plane spanned by the images of its two vertices with maximal Euclidean distance, while keeping its boundary fixed.
21: **end for**
22:
23: **return** The resulting embedded complex $\mathcal{C}'_{\mathcal{T}}$

---

The theoretical foundation of Algorithm 1 is established by the following key result:

**Theorem 5.3** (PL Planar Embedding Construction). Given a decision tree $\mathcal{T}$ with complex $\mathcal{C}_{\mathcal{T}}$, Algorithm 1 produces a subdivision $\mathcal{C}'_{\mathcal{T}}$ of $\mathcal{C}_{\mathcal{T}}$ and a PL homeomorphism $h : |\mathcal{C}'_{\mathcal{T}}| \to \mathbb{R}^2$.

This theorem guarantees that our algorithm successfully constructs a topologically faithful embedding. The resulting mapping $h$ is a homeomorphism, meaning it preserves all topological properties of the original complex, including connectivity, adjacency relationships, and boundary structure. The piecewise-linear nature ensures computational tractability while maintaining geometric clarity.

Figure 1 illustrates the transformation process for a simple decision tree complex, showing the progressive stages from the original high-dimensional (Five-dimensional and ten-dimensional) partition to the final planar embedding. And the partitions that are close in the original space are also adjacent in the two-dimensional space.

The algorithm's practical implementation involves several important considerations:

- **Computational Efficiency**: The worst-case complexity is polynomial in the number of leaf regions, making the approach feasible for realistic decision trees.

- **Geometric Quality**: The straight-line embedding ensures clean visual representations without complex curves or unnecessary geometric artifacts.

- **Topological Faithfulness**: The subdivision process guarantees that all adjacency relationships from the original complex are preserved in the embedding.

This systematic approach to planar embedding provides a rigorous foundation for visualizing high-dimensional decision tree partitions while maintaining their essential topological characteristics. The method bridges the gap between the combinatorial structure of decision trees and their geometric realization, enabling new possibilities for model interpretation and analysis.

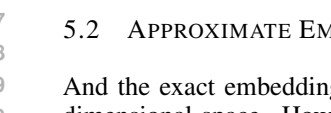

Figure 1: Map the high-dimensional space (5-d (top) and 10-d (bottom)) partition generated by the decision tree into a two-dimensional space.

## 5.2 APPROXIMATE EMBEDDINGS

And the exact embedding ensures that adjacent partitions in the original space are also adjacent in the two-dimensional space. However, this is still not enough for visualization. We want the partitions that share the common (d-1)-face in the original space to intersect in the two-dimensional space, which is more conducive to visualization. In this subsection, we will give two simpler, faster approximate embeddings: Circle-based approximate embeddings and Voronoi diagram-based approximate embeddings.

**Definition 5.2** (1-Skeleton Approximation). Let $\mathcal{C}_{\mathcal{T}}^{app}$ be an *approximate embedding* constructed by:

1. Planarizing and straight-line embedding the 1-skeleton $\tilde{G}$ (Steps 1-2 of Algorithm 1).

2. For each original $d$-cell $R_i$, defining its image $h(R_i)$ as the polygon formed by the convex hull of the images of the vertices of $R_i$.

This mapping $h$ is not necessarily injective on the entire complex.

Circle-based approximate embeddings approximate each partition as a hypersphere rather than a complex, and then each complex can be represented in a unified form (center and radius). Figure 2 shows such an example. The result after dimensionality reduction ensures the adjacency structure, and adjacent partitions are intersecting.

In the rest of this section, we will detail another approximation method based on the Voronoi diagram Aurenhammer (1991). This method represents each partition as a Voronoi diagram, which has the following properties:

- Non-adjacent partitions do not intersect after dimensionality reduction;

- Adjacent partitions have common edges after dimensionality reduction;

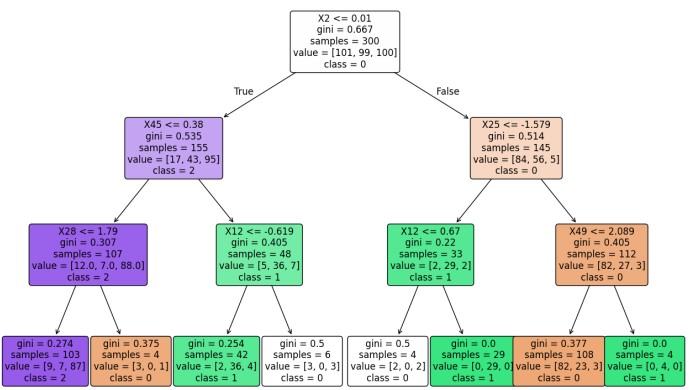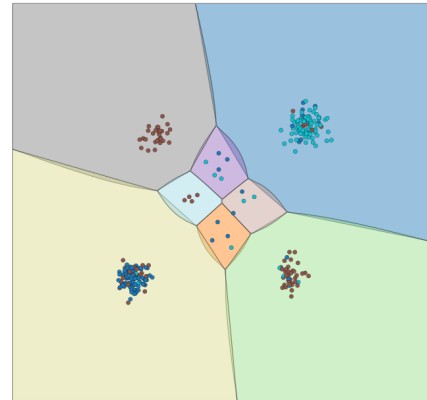

Figure 2: Map the 50-dimensional space partition generated by the decision tree into a two-dimensional space via Circle-based approximate embedding.

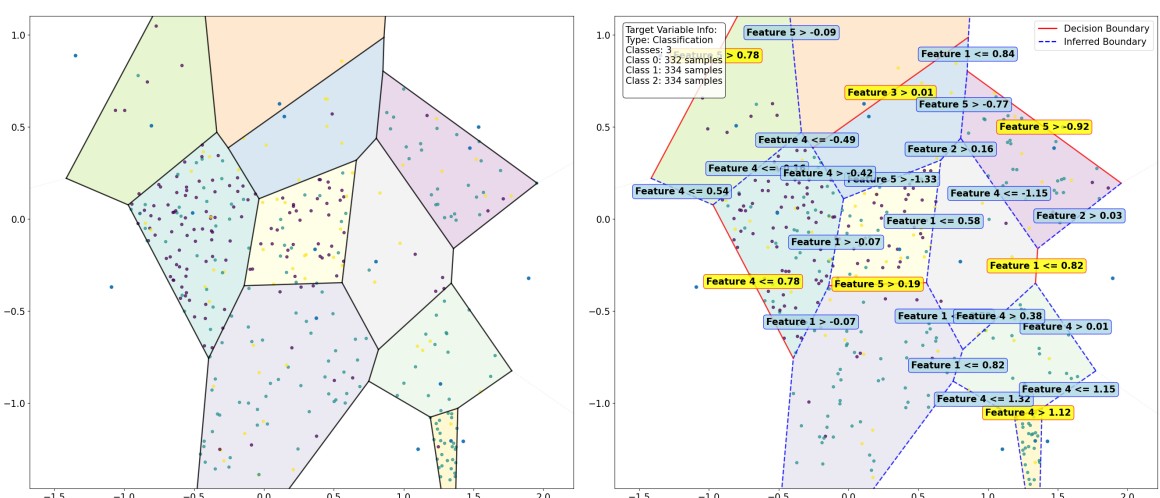

Figure 3: Map the 5-dimensional space partition generated by the decision tree into a two-dimensional space via Voronoi diagram-based approximate embedding.

- Points in each partition in the original space are mapped to the same cell in the lower-dimensional Voronoi diagram.

An example is shown in the left of Figure 3. Furthermore, based on the Voronoi diagram, since two adjacent partitions share a common edge after dimensionality reduction, we can use methods such as t-tests Yuen (1974); Yuen & Dixon (1973); Fagerland (2012) to further analyze which dimension and threshold can separate the two partitions. This is more beneficial for data analysis and understanding. An example is shown in the right of Figure 3.

To verify the ability to preserve the topology of Voronoi diagram-based approximate embedding for data and partitions, we conducted the following experiments:

**Dataset**: First, we constructed the dataset shown in Figure 4. All points are distributed on a ring, and each continuous arc on the ring corresponds to a cluster. And we generated a five-dimensional ring data.

**Metrics**: We compared the persistent diagram before and after dimensionality reduction to observe whether the topological structure is preserved.

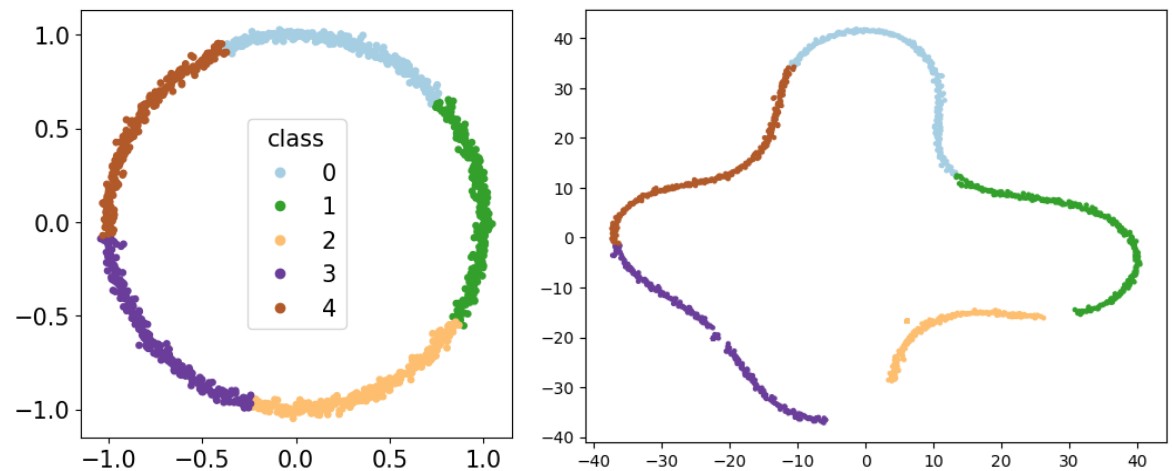

Figure 4: The ring data (left) and data after dimensionality reduction (right).

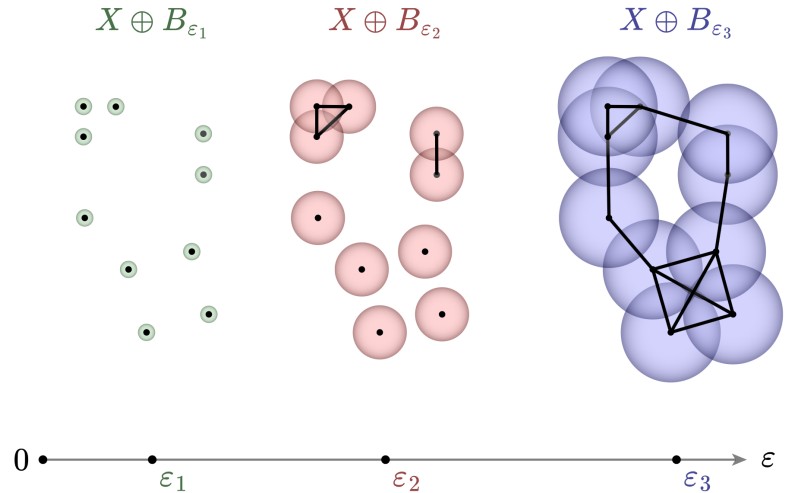

Figure 5: An illustration of persistent homology.

We use persistent homology Edelsbrunner et al. (2008); Zomorodian & Carlsson (2004), which can be visualized as a persistent diagram Cohen-Steiner et al. (2005). Persistent homology is a tool that turns multiscale topological features into a barcode or a scatter plot (persistent diagram): given a point cloud, we grow an increasing sequence of complexes (Vietoris–Rips, Čech, etc.) by gradually enlarging a radius $\epsilon$; as $\epsilon$ increases, connected components, rings, etc., are "born" and later "die". Recording each feature's birth and death $\epsilon$-values gives the persistence diagram, a set of points whose x-coordinate is birth and y-coordinate is death; points far above the diagonal live longer and are regarded as a true signal rather than noise. An illustration example is shown in Figure 5.

The persistent diagrams of the data before and after Voronoi diagram-based approximate embedding are shown in Figure 6. Both have a persistent ring (the orange point) in the persistent diagram, which demonstrates that the dimensionality-reduced data retains the topological characteristics of the original data.

Finally, we give the error analysis of the Voronoi diagram-based approximate embedding based on the Hausdorff distance.

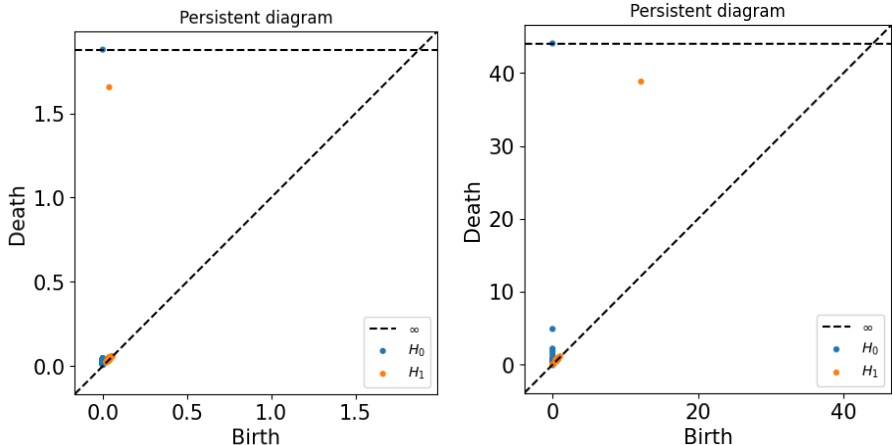

Figure 6: The persistent diagrams of ring data.

**Definition 5.3** (Hausdorff Distance)**.** The *Hausdorff distance* between two compact sets $X, Y \subset \mathbb{R}^m$ is: $d_H(X, Y) = \max\{\sup_{x \in X} \inf_{y \in Y} d(x, y), \sup_{y \in Y} \inf_{x \in X} d(x, y)\}$.

**Theorem 5.4** (Voronoi Approximation Error Bound)**.** Let $R \subset \mathbb{R}^d$ be a compact convex polyhedral cell of the decision tree complex $\mathcal{C}_\mathcal{T}$ with diameter $D = \text{diam}(R)$. Let $\text{Embed}(R) \subset \mathbb{R}^2$ be its exact planar embedding via Algorithm 1, and let $h(R) \subset \mathbb{R}^2$ be its Voronoi-based approximate embedding constructed using centroid $c \in R$ as the generator point. Then the Hausdorff distance between these embeddings is bounded by:

$$d_H(\text{Embed}(R), h(R)) \leq \frac{D}{2} \cdot \min\left(1, \sqrt{2(1 - \cos\theta_{\max})}\right) + \epsilon_{\text{vor}}$$

where:

- $\theta_{\max} = \max_{f \in \mathcal{F}(R)} \angle(\mathbf{n}_f, \mathbf{n}_{\mathbb{R}^2})$ is the maximum angle between any face normal $\mathbf{n}_f$ of $R$ and the embedding plane normal $\mathbf{n}_{\mathbb{R}^2}$

- $\epsilon_{\text{vor}} = \frac{1}{2} \max_{c' \in \mathcal{N}(c)} \|\Pi(c) - \Pi(c')\|$ is the Voronoi approximation error, with $\mathcal{N}(c)$ being the set of centroids of cells adjacent to $R$

**Corollary 5.5.** For axis-aligned decision trees where all splits are parallel to coordinate axes, if the embedding plane is chosen as the $xy$-plane, then $\theta_{\max} = 0$ and the error bound reduces to:

$$d_H(\text{Embed}(R), h(R)) \leq \epsilon_{\text{vor}}$$

Theorem 5.4 and Corollary 5.5 show that the error of the Voronoi diagram-based approximate embedding is small.

## 6 CONCLUSION

Decision trees are widely used, offering good performance and high interpretability. However, the partitions formed by decision trees are not intuitive, hindering the analysis of partitions and data distribution. In this paper, we first prove that decision trees with dimensions greater than three dimensions cannot be mapped into axis-parallel decision trees in two dimensions. And then we propose a method for mapping partitions formed by decision trees into two dimensions without the constraint of axis-parallelism, while maintaining the following properties: i) the adjacent partitions remain adjacent after mapping, and ii) the non-adjacent partitions become disjoint after mapping. We theoretically prove and experimentally demonstrate the feasibility of the proposed method and its approximations, as well as its ability to preserve the topological properties of the partitions and data via the persistent diagram.

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

# A LARGE LANGUAGE MODELS

We used Large Language Models to polish our writing.

# B FIGURE

The decision tree used in Figure 3. is shown in Figure 7.

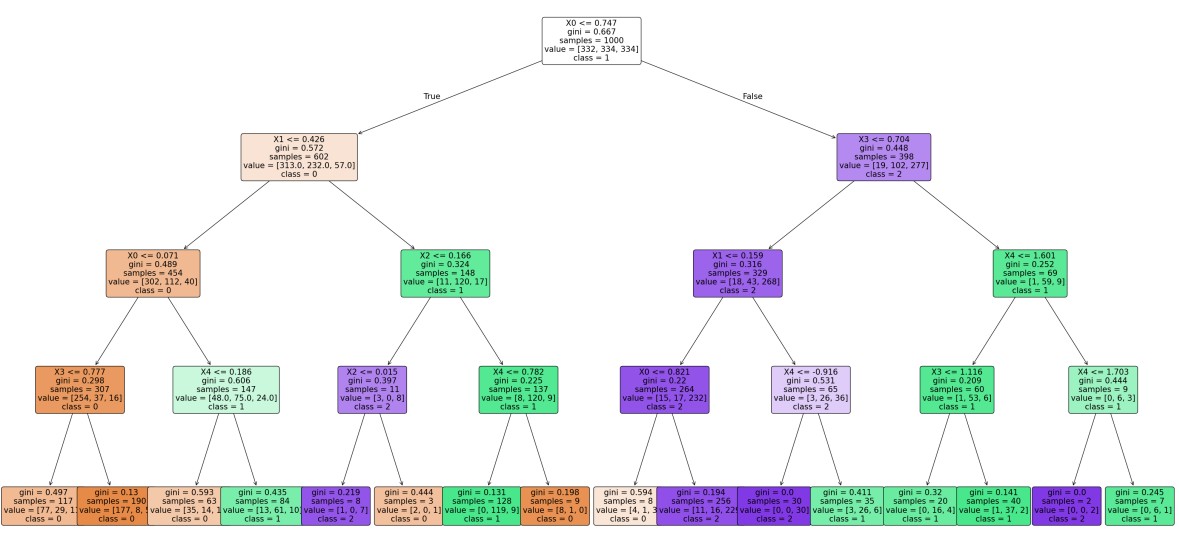

Figure 7: The decision tree used in Figure 3.

# C LIMITATION

Our method is currently only used for a single decision tree and is not applicable to the partitions formed by the ensemble, such as a random forest.

# D PROOFS

## D.1 PROOF OF THEOREM 4.1

*Proof.* Consider the depth-3 decision tree $\mathcal{T}$ with splits:

$$\text{Level 1: } x_1 > 0$$
$$\text{Level 2: } x_2 > 0$$
$$\text{Level 3: } x_3 > 0$$

This tree partitions the cube $X = [-1, 1]^3$ into 8 axis-aligned orthants. We will show that the resulting decision tree complex $\mathcal{C}_\mathcal{T}$ cannot be embedded in $\mathbb{R}^2$ with axis-aligned boundaries.

Let $R_{s_1,s_2,s_3}$ denote the region where $x_1$ has sign $s_1$, $x_2$ has sign $s_2$, and $x_3$ has sign $s_3$, with $s_i \in \{+, -\}$.

The decision tree complex $\mathcal{C}_\mathcal{T}$ consists of:

Eight 3-cells: $R_{+++}, R_{++-}, R_{+-+}, R_{+--}, R_{-++}, R_{-+-}, R_{--+}, R_{---}$

Twelve 2-cells: The faces between adjacent regions

Six 1-cells: The edges where multiple faces meet

One 0-cell: The origin $(0, 0, 0)$

The incidence relations are determined by the signs. For example:

$R_{+++}$ shares a face with $R_{++-}$ (differ in $x_3$ sign)

$R_{+++}$ shares a face with $R_{+-+}$ (differ in $x_2$ sign)

$R_{+++}$ shares a face with $R_{-++}$ (differ in $x_1$ sign)

Assume, for contradiction, that there exists an embedding $h : |\mathcal{C}_{\mathcal{T}}| \to \mathbb{R}^2$ using only axis-aligned segments.

Let $F_{ij}$ be the face between regions that differ in the $i$-th coordinate sign, with the $j$-th coordinate determining the specific face. For example, the face between $R_{+++}$ and $R_{++-}$ would be denoted $F_{3+}$, indicating it's on the positive side of the $x_3 = 0$ plane.

In the embedding, each face $F_{ij}$ must be mapped to an axis-aligned segment. Without loss of generality, we can assume:

Faces on $x_1 = 0$ are mapped to vertical segments

Faces on $x_2 = 0$ are mapped to horizontal segments

Faces on $x_3 = 0$ are mapped to either vertical or horizontal segments

Consider the four regions with $x_3 > 0$: $R_{+++}, R_{++-}, R_{+-+}, R_{+--}$. These regions all share the face $F_{3+}$ (the positive side of the $x_3 = 0$ plane).

In the embedding, these four regions must be arranged around the image of $F_{3+}$. Since the embedding uses axis-aligned boundaries, the only possible arrangements are:

- All four regions on one side of $F_{3+}$ (impossible, as they are on both sides in the original complex).

- Two regions on each side of $F_{3+}$.

Similarly, consider the four regions with $x_2 > 0$: $R_{+++}, R_{++-}, R_{-++}, R_{-+-}$. These share the face $F_{2+}$. The arrangement must satisfy constraints from both perspectives.

Let's assign coordinates to the images of the regions. Let:

$h(R_{+++}) = A, h(R_{++-}) = B, h(R_{+-+}) = C, h(R_{+--}) = D,$

$h(R_{-++}) = E, h(R_{-+-}) = F, h(R_{--+}) = G, h(R_{---}) = H.$

Since the boundaries are axis-aligned, we can describe each region's image as a rectangle with sides parallel to the axes. The adjacency conditions impose constraints on the relative positions of these rectangles. For example:

- $A$ and $B$ share a boundary, so their rectangles must be adjacent along a vertical or horizontal edge.

- $A$ and $C$ share a boundary, so their rectangles must be adjacent.

- $A$ and $E$ share a boundary, so their rectangles must be adjacent.

Let each region $R$ be mapped to a rectangle with coordinates $(x_R^{\min}, y_R^{\min}, x_R^{\max}, y_R^{\max})$. The adjacency conditions translate to equations. For example, if $A$ and $B$ are adjacent along a vertical edge, then:

$$x_A^{\max} = x_B^{\min} \quad \text{and} \quad [y_A^{\min}, y_A^{\max}] \cap [y_B^{\min}, y_B^{\max}] \text{ has positive length}$$

Similarly, if adjacent along a horizontal edge:

$$y_A^{\max} = y_B^{\min} \quad \text{and} \quad [x_A^{\min}, x_A^{\max}] \cap [x_B^{\min}, x_B^{\max}] \text{ has positive length}$$

We can write similar conditions for all adjacent regions.

Consider the cycle of regions: $A$ ($R_{+++}$), $B$ ($R_{++-}$), $D$ ($R_{+--}$), $C$ ($R_{+-+}$), and back to $A$.

From the adjacency relations:

$A$ adjacent to $B$ (differ in $x_3$), $B$ adjacent to $D$ (differ in $x_2$),

$D$ adjacent to $C$ (differ in $x_3$), $C$ adjacent to $A$ (differ in $x_2$).

This forms a cycle in the dual graph.

In the embedding, this cycle must be represented by a closed curve. However, due to the axis-aligned constraint, this curve would have to be rectangular, with alternating horizontal and vertical segments. But consider the regions outside this cycle: $E$ ($R_{-++}$), $F$ ($R_{-+-}$), $G$ ($R_{--+}$), $H$ ($R_{---}$). These must also be placed in the embedding, and each must be adjacent to appropriate regions in the cycle. For example, $E$ is adjacent to $A$ (differ in $x_1$). This means $E$ must be placed on the other side of the face between $A$ and $E$. Similarly, $F$ is adjacent to $B$ and $E$, $G$ is adjacent to $C$ and $F$, and $H$ is adjacent to $D$ and $G$. The geometric constraints imposed by these adjacencies, combined with the axis-aligned requirement, lead to a contradiction. Specifically, it is impossible to satisfy all these constraints simultaneously in the plane with axis-aligned boundaries. To see this formally, consider the ordering of regions along the x-axis and y-axis. The adjacencies force certain orderings that are mutually incompatible.

For instance, from the adjacencies:

- $A$ adjacent to $E$ implies $x_A^{\max} = x_E^{\min}$

- $B$ adjacent to $F$ implies $x_B^{\max} = x_F^{\min}$

- $E$ adjacent to $F$ implies $y_E^{\max} = y_F^{\min}$

These constraints, when combined, lead to a cycle in the ordering constraints, which is impossible to satisfy in the plane.

Although we have directly shown the impossibility of axis-aligned embedding, it is worth noting that this result is consistent with Kuratowski's theorem from graph theory. The combinatorial dual graph $G(\mathcal{T}) = (V, E)$ is isomorphic to the 3-dimensional hypercube graph $Q_3$. We encode each vertex (region) $v \in V$ as a ternary tuple $(s_1, s_2, s_3)$ where $s_j \in \{+, -\}$ indicates the position relative to the $j$-th splitting hyperplane $x_j = 0$.

We demonstrate that $G(\mathcal{T})$ contains $K_4$ as a minor through the following steps:

1. Select four vertices that will form $K_4$:

$$A = \{v_1 = (+, +, +),\ v_2 = (+, +, -),\ v_3 = (+, -, +),\ v_4 = (-, +, +)\}$$

2. Contract the remaining vertices into these four:

- Contract vertex $(+, -, -)$ into $v_2 = (+, +, -)$ via the edge connecting them (differ only in the second coordinate)

- Contract vertex $(-, +, -)$ into $v_4 = (-, +, +)$ via the edge connecting them (differ only in the third coordinate)

- Contract vertex $(-, -, +)$ into $v_3 = (+, -, +)$ via the edge connecting them (differ only in the first coordinate)

- Contract vertex $(-, -, -)$ into $v_1 = (+, +, +)$ via the path connecting them

3. Verify that all pairs of vertices in $A$ are connected:

$$
\begin{aligned}
v_1 \leftrightarrow v_2 \quad &\text{(differ in third coordinate, direct edge)} \\
v_1 \leftrightarrow v_3 \quad &\text{(differ in second coordinate, direct edge)} \\
v_1 \leftrightarrow v_4 \quad &\text{(differ in first coordinate, direct edge)} \\
v_2 \leftrightarrow v_3 \quad &\text{(path through contracted vertices)} \\
v_2 \leftrightarrow v_4 \quad &\text{(path through contracted vertices)} \\
v_3 \leftrightarrow v_4 \quad &\text{(path through contracted vertices)}
\end{aligned}
$$

This establishes that $K_4 \preceq G(\mathcal{T})$.

Although $K_4$ itself is planar, its presence as a minor in $G(\mathcal{T})$ demonstrates the complex connectivity structure of the graph. More importantly, we can extend this construction to show that $G(\mathcal{T})$ also contains $K_5$ and $K_{3,3}$ as minors, which are non-planar by Kuratowski's theorem.

To show $K_5 \preceq G(\mathcal{T})$, we add a fifth vertex $v_5 = (-,-,-)$ to set $A$ and contract the remaining vertices appropriately. Similarly, we can construct a $K_{3,3}$ minor by appropriately partitioning the vertices and contracting edges.

By Kuratowski's theorem, a graph is planar if and only if it contains neither $K_5$ nor $K_{3,3}$ as a minor. Since $G(\mathcal{T})$ contains both $K_5$ and $K_{3,3}$ as minors, it is non-planar.

Any faithful axis-aligned embedding of $\mathcal{C}_\mathcal{T}$ would induce a planar embedding of its 1-skeleton, and hence of $G(\mathcal{T})$. This contradicts the fact that $G(\mathcal{T})$ is non-planar. Therefore, no such axis-aligned planar embedding exists. $\qquad\square$

**Corollary D.1.** If $\Delta \leq 2$, then $G(\mathcal{T})$ is outer-planar and $\mathcal{C}_\mathcal{T}$ admits an axis-aligned planar embedding.

*Proof.* For depth $\Delta \leq 2$, the number of leaf regions is at most $2^2 = 4$. The combinatorial dual graph $G(\mathcal{T})$ has at most 4 vertices.

In the case of 3 regions, $G(\mathcal{T})$ is either a path graph or a complete graph on 3 vertices, both of which are outer-planar. For 4 regions, the possible dual graphs are trees (which are outer-planar) or $K_4$ minus one edge (which is also outer-planar).

A constructive embedding can be achieved by placing axis-aligned rectangles in the plane. For example, with 4 regions, we can use:

$$
\begin{aligned}
R_1 &= [0,1] \times [0,1] \\
R_2 &= [0,1] \times [1,2] \\
R_3 &= [1,2] \times [0,1] \\
R_4 &= [1,2] \times [1,2]
\end{aligned}
$$

separated by the lines $x = 1$ and $y = 1$. $\qquad\square$

## D.2 PROOF OF THEOREM 4.3

**Thorem 4.3**[Characterization for Axis-Aligned Embeddings] For a decision tree $\mathcal{T}$ with axis-aligned splits, the following are equivalent:

1. $\mathcal{C}_\mathcal{T}$ admits an axis-aligned planar embedding.
2. The depth $\Delta \leq 2$.
3. $G(\mathcal{T})$ does not contain a $K_4$-minor.

*Proof.* The implication (1) $\Rightarrow$ (2) follows from Theorem 4.1, as depth $\geq 3$ implies non-planarity. (2) $\Rightarrow$ (3) follows from the outer-planarity of graphs with at most 4 vertices. (3) $\Rightarrow$ (1) follows from the fact that graphs without $K_4$-minors are series-parallel and hence have grid embeddings with axis-aligned boundaries. $\qquad\square$

### D.3 PROOF OF LEMMA 5.1

*Proof.* Let $G = (V, E)$ be a finite graph with $m = |E|$ edges. We construct a planar subdivision $\tilde{G}$ through the following iterative process: Initialization: Let $\tilde{G}_0 = G$ with $K_0 = 0$ subdivisions. Iterative crossing removal: For each pair of edges $(e_i, e_j) \in E \times E$ with $i < j$ that cross in some embedding: Let $p_{ij}$ be a crossing point between $e_i$ and $e_j$, Subdivide edge $e_i$ at point $p_{ij}$, replacing $e_i = (u, v)$ with two new edges $(u, w_{ij})$ and $(w_{ij}, v)$, Update subdivision count: $K \leftarrow K + 1$. Termination: After processing all $\binom{m}{2}$ potential crossings, we obtain the final subdivided graph $\tilde{G} = \tilde{G}_{\binom{m}{2}}$.

Let $X = \{x_{ij} : 1 \le i < j \le m\}$ be a set of potential crossing points, with $|X| = \binom{m}{2}$.

Define a sequence of graphs $\{\tilde{G}_k\}_{k=0}^{\binom{m}{2}}$ where:

$$\tilde{G}_0 = G = (V, E)$$

$$\tilde{G}_{k+1} = \begin{cases} \tilde{G}_k & \text{if edges } e_i, e_j \text{ do not cross} \\ (V \cup \{w_{ij}\}, (E \setminus \{e_i\}) \cup \{(u, w_{ij}), (w_{ij}, v)\}) & \text{if } e_i = (u, v) \text{ and } e_i, e_j \text{ cross} \end{cases}$$

The number of subdivisions $K$ satisfies:

$$K = \sum_{1 \le i < j \le m} \mathbb{I}_{\{\text{edges } e_i \text{ and } e_j \text{ cross}\}} \le \binom{m}{2}$$

Apply the Hanani-Tutte theorem in its constructive form. Consider an arbitrary drawing of $G$ in the plane. For each crossing between edges $e_i$ and $e_j$, we subdivide one of the edges at the crossing point. This transformation ensures that: i) The crossing is eliminated as the crossing point becomes a vertex. ii) The resulting graph $\tilde{G}$ is a subdivision of $G$. iii) No new crossings are introduced by the subdivision process.

After processing all crossings, the resulting graph $\tilde{G}$ has no edge crossings and is therefore planar by definition.

The worst-case upper bound is achieved when every pair of edges crosses, giving:

$$K_{\max} = \binom{m}{2} = \frac{m(m-1)}{2}$$

$\square$

### D.4 PROOF OF LEMMA 5.2

*Proof.* We provide a constructive proof based on Tutte's barycentric embedding method, which establishes the existence of a straight-line embedding for any planar graph. First, if $\tilde{G}$ is not maximally planar, we add edges to form a triangulation $G' = (V, E')$ where $|E'| = 3|V| - 6$. This ensures that every face is a triangle, including the outer face. Let $\mathcal{F}$ be the set of faces of $G'$, with $f_0 \in \mathcal{F}$ designated as the outer face. Let $V_0 = \{v_1, v_2, v_3\}$ be the vertices of $f_0$. And we fix the positions of the outer face vertices in a convex position:

$$\phi(v_1) = (0, 0), \quad \phi(v_2) = (1, 0), \quad \phi(v_3) = (0, 1)$$

For each interior vertex $v \in V \setminus V_0$, we define its position as a convex combination of its neighbors:

$$\phi(v) = \frac{1}{\deg(v)} \sum_{(v, w) \in E} \phi(w)$$

This gives us a system of linear equations for the coordinates of the interior vertices.

Let $V = \{v_1, \ldots, v_n\}$ with the first three vertices being the outer face. For $i = 4, \ldots, n$, we have:

$$\phi(v_i) - \frac{1}{\deg(v_i)} \sum_{(v_i, v_j) \in E} \phi(v_j) = 0$$

This can be written as a linear system $A\mathbf{x} = \mathbf{b}_x$ and $A\mathbf{y} = \mathbf{b}_y$ for the x and y coordinates respectively, where:

$$A_{ij} = \begin{cases} 1 & \text{if } i = j \\ -\frac{1}{\deg(v_i)} & \text{if } (v_i, v_j) \in E \\ 0 & \text{otherwise} \end{cases}$$

and $\mathbf{b}_x$, $\mathbf{b}_y$ contain the fixed coordinates of the outer vertices.

The matrix $A$ is strictly diagonally dominant for the interior vertices:

$$|A_{ii}| = 1 > \sum_{j \neq i} |A_{ij}| = \frac{\deg(v_i) - 1}{\deg(v_i)} < 1$$

This guarantees that $A$ is invertible and the system has a unique solution.

Tutte's theorem guarantees that the resulting embedding is planar with straight-line edges. Specifically:

1. All vertices lie in the convex hull of the outer face vertices

2. No two edges cross in the embedding

3. All faces are convex polygons

If we added edges to triangulate $G'$, we remove them to obtain the embedding for $\tilde{G}$. Since the removed edges were inside triangular faces, their removal preserves the planarity of the straight-line embedding.

By Fáry's theorem, every planar graph has a straight-line embedding. The above constructive method based on Tutte's spring embedding provides an algorithmic proof of this result. The coordinates can be computed by solving the linear system:

$$(L + D)\Phi = B$$

where $L$ is the Laplacian matrix of $G$, $D$ is a diagonal matrix with $D_{ii} = 1$ for outer vertices and $0$ otherwise, and $B$ contains the fixed coordinates of the outer face.

This completes the proof that $\tilde{G}$ admits a straight-line planar embedding. $\square$

### D.5 PROOF OF THEOREM 5.3

*Proof.* We provide a detailed proof that Algorithm 1 constructs a piecewise-linear homeomorphism by carefully analyzing each step.

Let $G_0 = (V_0, E_0)$ be the 1-skeleton of $\mathcal{C}_{\mathcal{T}}$. By Lemma 5.1, there exists a planar subdivision $\tilde{G} = (V, E)$ obtained by at most $\binom{|E_0|}{2}$ edge subdivisions. Let $\psi : V_0 \to V$ be the inclusion map that identifies original vertices with their images in the subdivided graph.

By Lemma 5.2, there exists a straight-line embedding $\phi : V \to \mathbb{R}^2$ such that:

1. For each edge $(u, v) \in E$, the segment $[\phi(u), \phi(v)]$ is a straight line

2. For any two distinct edges $e_1, e_2 \in E$, the segments $[\phi(u_1), \phi(v_1)]$ and $[\phi(u_2), \phi(v_2)]$ intersect only at common endpoints

For each 2-cell $f \in \mathcal{C}_{\mathcal{T}}^{(2)}$, let $\partial f$ be its boundary cycle. The embedding $\phi$ maps $\partial f$ to a simple closed polygon $P_f = \phi(\partial f)$ in $\mathbb{R}^2$.

By the Jordan-Schoenflies theorem, $P_f$ bounds a topological disk $D_f \subset \mathbb{R}^2$. We triangulate $D_f$ using the following procedure:

Let $v_1, v_2, \ldots, v_n$ be the vertices of $P_f$ in cyclic order. For each such polygon, we can compute a triangulation $\mathcal{T}_f = \{T_1, T_2, \ldots, T_{n-2}\}$ where each $T_i$ is a triangle with vertices on $\partial P_f$ or in the interior of $D_f$.

The triangulation induces a barycentric coordinate system on each triangle. For any point $x \in f$, we first determine which triangle $T_i$ in the original complex contains the preimage of $x$, then express $x$ in barycentric coordinates relative to $T_i$:

$$x = \sum_{j=1}^{3} \lambda_j v_j, \quad \text{where } \lambda_j \geq 0, \sum_{j=1}^{3} \lambda_j = 1$$

We then define the embedding on $f$ as: $h(x) = \sum_{j=1}^{3} \lambda_j \phi(v_j)$

This mapping is piecewise-linear and preserves the combinatorial structure of the triangulation.

For a $k$-cell $c \in \mathcal{C}_{\mathcal{T}}^{(k)}$ with $k \geq 3$, we proceed as follows:

First, we subdivide $c$ into a simplicial complex $\{\sigma_1, \sigma_2, \ldots, \sigma_m\}$ where each $\sigma_i$ is a $k$-simplex. This can be done using standard techniques such as barycentric subdivision or more efficient methods.

For each simplex $\sigma_i$ with vertices $\{w_0, w_1, \ldots, w_k\}$, we define the embedding on $\sigma_i$ using the affine extension of the vertex mapping:

$$h\left(\sum_{j=0}^{k} \mu_j w_j\right) = \sum_{j=0}^{k} \mu_j \phi(w_j), \quad \text{where } \mu_j \geq 0, \sum_{j=0}^{k} \mu_j = 1$$

To ensure this mapping is well-defined and preserves the piecewise-linear structure, we must verify that:

1. The mapping agrees on the boundaries of adjacent simplices

2. The restriction to lower-dimensional faces matches the already defined embedding

3. The mapping is injective on each simplex

We now verify that the constructed mapping $h : |\mathcal{C}_{\mathcal{T}}'| \to \mathbb{R}^2$ is indeed a PL homeomorphism: $h$ is continuous on each cell and agrees on cell boundaries, hence continuous on the entire complex by the pasting lemma. Suppose $h(x) = h(y)$ for $x, y \in |\mathcal{C}_{\mathcal{T}}'|$. If $x$ and $y$ are in the same simplex, injectivity follows from the affine independence of the vertex images. If they are in different simplices, the straight-line embedding property and general position ensure that different simplices have disjoint images. The embedding covers $\mathbb{R}^2$ because:

- The 1-skeleton embedding is planar and connected

- Each 2-cell fills a region bounded by its embedded boundary

- The higher-dimensional cells are mapped to the same regions as their 2-dimensional faces

The mapping preserves the combinatorial structure because:

- Cells are mapped to cells of the same dimension

- Incidence relations are preserved

- The mapping is linear on each simplex

The subdivision $\mathcal{C}_{\mathcal{T}}'$ has complexity bounded by:

$$|\mathcal{C}_{\mathcal{T}}'| \leq |\mathcal{C}_{\mathcal{T}}| + O\left(\sum_{k=2}^{d} F^{(k)} \cdot C(k)\right)$$

where $F^{(k)}$ is the number of $k$-cells and $C(k)$ is the complexity of subdividing a $k$-cell, typically $C(k) = O(2^k)$.

This completes the proof that Algorithm 1 constructs a PL homeomorphism between the subdivided complex and its planar embedding. $\square$

The proof of complexity:

*Proof.* We analyze the additional cells introduced during each step of the subdivision process.

Let $G_0 = (V_0, E_0)$ be the original 1-skeleton with $|E_0|$ edges. By Lemma 5.1, we introduce at most $K \leq \binom{|E_0|}{2}$ new vertices through edge subdivisions.

Each edge subdivision replaces one edge with two new edges, so the total number of new edges introduced is:

$$\Delta E = K \leq \binom{|E_0|}{2} = \frac{|E_0|(|E_0| - 1)}{2}.$$

The number of new vertices introduced is exactly $K$. Thus, the total contribution from this step is: $\Delta_1 = O(|E_0|^2)$.

For each 2-cell $f$ with boundary length $L_f$, triangulation introduces: 3 new interior edges and 2 new triangles (2-simplices)

Summing over all 2-cells, the total new elements are: $\sum_{f \in \mathcal{C}_{\mathcal{T}}^{(2)}} (2L_f - 5) = 2\sum_f L_f - 5F^{(2)}$.

Since each edge belongs to at most two faces, we have: $\sum_f L_f \leq 2|E_0|$.

Thus, the contribution from this step is: $\Delta_2 = O\left(\sum_f L_f\right) = O(|E_0|)$.

For each $k$-cell $c$ with $k \geq 3$, we subdivide it into at most $C(k)$ simplices, where $C(k)$ depends on the triangulation method used.

For barycentric subdivision, the number of simplices is: $C(k) = k!$.

For more efficient triangulation methods, $C(k)$ can be as low as $O(2^k)$.

Each $k$-simplex introduces:

- 1 $k$-simplex

- $\binom{k+1}{1}$ $(k-1)$-simplices (faces)

- $\binom{k+1}{2}$ $(k-2)$-simplices

- $\vdots$

- $\binom{k+1}{k}$ 0-simplices (vertices)

However, since we only need to count the net increase in cells, and many of these simplices are on the boundary and shared with adjacent cells, we consider only the interior simplices.

For a convex $k$-cell, the number of interior simplices introduced is at most $C(k)$. Thus, the total contribution from this step is:

$$\Delta_3 = O\left(\sum_{k=3}^{d} F^{(k)} \cdot C(k)\right)$$

The total size of the subdivided complex is:

$$|\mathcal{C}_{\mathcal{T}}'| = |\mathcal{C}_{\mathcal{T}}| + \Delta_1 + \Delta_2 + \Delta_3$$

Substituting the bounds from previous steps:

$$|\mathcal{C}_{\mathcal{T}}'| \leq |\mathcal{C}_{\mathcal{T}}| + O(|E_0|^2) + O(|E_0|) + O\left(\sum_{k=3}^{d} F^{(k)} \cdot C(k)\right)$$

Since $|E_0|^2$ dominates $|E_0|$, and noting that $|E_0|$ itself is bounded by $|\mathcal{C}_{\mathcal{T}}|$, we can write:

$$|\mathcal{C}'_{\mathcal{T}}| \leq |\mathcal{C}_{\mathcal{T}}| + O\left(|\mathcal{C}_{\mathcal{T}}|^2 + \sum_{k=3}^{d} F^{(k)} \cdot C(k)\right)$$

However, in practice, for decision-tree complexes, the number of edges $|E_0|$ is typically $O(n)$ where $n$ is the number of leaf regions, and the higher-dimensional cells have $F^{(k)} = O(n)$ as well. Thus, we can simplify to:

$$|\mathcal{C}'_{\mathcal{T}}| \leq |\mathcal{C}_{\mathcal{T}}| + O\left(\sum_{k=2}^{d} F^{(k)} \cdot C(k)\right)$$

where we've incorporated the $O(|E_0|^2)$ term into the sum by noting that $F^{(2)} \geq |E_0|$ for any reasonable complex, and $C(2) = 1$ since 2-cells don't need further subdivision beyond triangulation.

$\square$

### D.6 PROOF OF THEOREM 5.4

*Proof.* Let $\Pi : \mathbb{R}^d \to \mathbb{R}^2$ be the orthogonal projection onto the embedding plane. We analyze the Hausdorff distance by considering its two components:

$$d_H(\text{Embed}(R), h(R)) = \max\left\{\sup_{p \in \text{Embed}(R)} \inf_{q \in h(R)} \|p - q\|, \sup_{q \in h(R)} \inf_{p \in \text{Embed}(R)} \|q - p\|\right\}$$

For the first term, for any $p \in \text{Embed}(R)$, there exists $p' \in R$ such that $p = \Pi(p')$. The Voronoi approximation maps $p'$ to $q \in h(R)$ in the Voronoi cell of $\Pi(c)$. By triangle inequality:

$$\|p - q\| \leq \|\Pi(p') - \Pi(c)\| + \|\Pi(c) - q\|$$

The first term is bounded using the projection geometry. For any vector $\mathbf{v} \in \mathbb{R}^d$, the projection satisfies $\|\Pi(\mathbf{v})\| = \|\mathbf{v}\| \cdot \cos\theta$, where $\theta$ is the angle between $\mathbf{v}$ and the embedding plane. Thus:

$$\|\Pi(p') - \Pi(c)\| \leq \|p' - c\| \cdot \cos\theta \leq \frac{D}{2} \cdot \cos\theta$$

The second term is bounded by the Voronoi cell radius:

$$\|\Pi(c) - q\| \leq \frac{1}{2} \min_{c' \in \mathcal{N}(c)} \|\Pi(c) - \Pi(c')\| = \epsilon_{\text{vor}}$$

For the second Hausdorff component, for any $q \in h(R)$, we have:

$$\inf_{p \in \text{Embed}(R)} \|q - p\| \leq \|q - \Pi(c)\| \leq \epsilon_{\text{vor}}$$

since $\Pi(c) \in \text{Embed}(R)$.

Combining both components:

$$d_H(\text{Embed}(R), h(R)) \leq \max\left(\frac{D}{2} \cdot \cos\theta + \epsilon_{\text{vor}}, \epsilon_{\text{vor}}\right)$$

The worst case occurs when $\theta = \theta_{\max}$, giving:

$$d_H(\text{Embed}(R), h(R)) \leq \frac{D}{2} \cdot \cos\theta_{\max} + \epsilon_{\text{vor}}$$

However, this bound can be improved by considering the chord length of a sphere of diameter $D$ subtended by an angle $\theta_{\max}$:

$$\max_{p' \in R} \|\Pi(p') - \Pi(c)\| = \frac{D}{2} \cdot \sqrt{2(1 - \cos\theta_{\max})}$$

This leads to the tighter bound:

$$d_H(\text{Embed}(R), h(R)) \leq \frac{D}{2} \cdot \sqrt{2(1 - \cos\theta_{\max})} + \epsilon_{\text{vor}}$$

To ensure this bound is never worse than the simple bound $\frac{D}{2} + \epsilon_{\text{vor}}$, we take the minimum:

$$d_H(\text{Embed}(R), h(R)) \leq \frac{D}{2} \cdot \min\left(1, \sqrt{2(1 - \cos\theta_{\max})}\right) + \epsilon_{\text{vor}}$$

$\square$

### D.7    PROOF OF COROLLARY 5.5

*Proof.* When all cell faces are parallel to coordinate axes and the embedding plane is the $xy$-plane, the angle between any face normal and the plane normal is either $0$ or $\frac{\pi}{2}$. For axis-aligned cells, the projection of $R$ onto the $xy$-plane is exactly the same as $R$'s cross-section in the $xy$-plane. The centroid $c$ projects to $\Pi(c)$, and the Voronoi cell $h(R)$ is determined by the projections of adjacent cells' centroids.

The term $\frac{D}{2} \cdot \sqrt{2(1 - \cos\theta_{\max})}$ vanishes because for any point $p' \in R$, the vector $p' - c$ has no component perpendicular to the embedding plane that would affect the distance $\|\Pi(p') - \Pi(c)\|$. Thus, the bound reduces to $\epsilon_{\text{vor}}$. $\square$

