# OpenReview forum: "Planar Homeomorphic Embeddings of Decision Tree"
_ICLR.cc/2026/Conference — ICLR 2026 Conference Withdrawn Submission_

### Official Review · Reviewer_QcQi · 2025-10-28

**Soundness:** 1
**Presentation:** 1
**Contribution:** 1
**Rating:** 0
**Confidence:** 5

**Summary:**

The method introduces a method for embedding data partitions induced by decision trees to two dimensions.

Its main method extracts the 1-skeleton of the polyhedral complex defined by combining the data separation on all levels of the decision tree $\mathcal{T}$, projects it in some way to $\mathbb{R}^2$ and planarizes it by adding new vertices on the intersections of edges. The embedding is further processed to get a 2D graph whose edges are straight lines. The generated embedding is the used to project the cells of $\mathcal{C}_{\mathcal{T}}$. Both points and cells are used for visualization.

Finally, two additional ways of embedding the $\mathcal{C}_{\mathcal{T}}$ cells are introduced in order to form adjacent regions. The first one approximates partitions with hyperspheres rather than complexes while the other one by mapping the cell partition to a Voronoi diagram.

**Strengths:**

- The paper has a relatively clean notation on the definition of the decision tree complex.

**Weaknesses:**

- Lemma 5.1 which describes a process of "planarization via subdivision" is completely flawed. It suggest to planarize graphs by using an initial projection to $\mathbb{R}^2$ and then adding all edge intersections as new nodes and the resulting part as new edges. This completely destroys any topological information of the original graph and is practically meaningless. To get a feeling of what can go wrong, imagine the 1-skeleton of $\mathcal{C}_{\mathcal{T}}$ in a random way, or even worse in a way which is deliberately bad by pushing away neighboring nodes. Then using the planarization method one could construct a planar graph which can be used to apply the rest of the algorithm. The final embedding would be valid according to the method, but would preserve no adjacency of the original space.

- Lemma 5.1 is mathematically wrong. My previous comment was based on the proof of the lemma in the appendix which makes it clear how the planarization is performed. The lemma states though that this process is possible by applying K many edge subdivisions. This is not possible, as e.g. take the counter-example used for theorem 4.3 where the 1-skeleton contains $K_5$ and $K_{3, 3}$ as minors. Any graph which is obtained by subdivisions of the 1-skeleton will contain it a minor (just contract one of the edges of each subdivision), thus will also contain the other two forbidden graphs as minors meaning it is not planar. It important to state that consecutive subdivisions cannot share a newly introduced node, like in the proof of 5.1. The main reason for not allowing this is precisely that such an operation can quickly destroy properties of a graph, including any topological information (s.a. the genus of a graph, in this case it can be reduced to 0) and graph complexity (s.a. the coloring number, n-connectivity).

- The initial embedding to $\mathbb{R}^2$, before making the graph planar, is never described.

- There are multiple parts of the paper where there are very little or no details on how they function. Those include the embedding of 2- and higher-dimensional cells to $\mathbb{R}^2$ (steps 2, 3 of the algorithm) and the two approximate embeddings introduced in section 5.2. The most striking example is the circle-based embedding which is superficially described in three lines.

- There is practically no experimental evidence to support the effectiveness of the method. There are multiple metrics which could measure how well the method can capture the level of adjacency or topology preservation. The example with the circle projected from $\mathbb{R}^2$ to $\mathbb{R}^2$ is trivial and does not provide any evidence. One would need more complicated topological structures for such evaluations, which of course would also show the failures of the method as it is not possible to fully preserve adjacency.

- The tree embedding examples shown are tiny toy examples which cannot demonstrate the usefulness of the method.

- The paper mentions a codebase at https://anonymous.4open.science/r/PLPE which is leads to an empty repository, so no hope of getting the missing details from the code.

**Questions:**

-

**Details Of Ethics Concerns:**

The paper aside from being of very poor quality contains statements which are obviously mathematically wrong, specifically theorem 5.1. At the same time there are parts of the proofs in the appendix where one gets the impression that the writers should have more than enough knowledge of graph theory to spot those issues. This leads me to believe that the paper is generated by an LLM or, worse, deliberately uses false statements in the hope of publishing or using the reviewers feedback to further develop the paper.

---

### Official Review · Reviewer_AzBb · 2025-10-31

**Soundness:** 2
**Presentation:** 2
**Contribution:** 2
**Rating:** 2
**Confidence:** 3

**Summary:**

This paper proposes a theoretically motivated framework for embedding the partition structure of a decision tree into two dimensions, treating leaf regions (from axis-aligned or general piecewise-linear splits) as the units to embed.
The key goal is to produce a planar, adjacency-preserving visualization of the tree’s decision regions by modeling the tree as a polyhedral complex and constructing a piecewise-linear (PL) embedding that maintains the combinatorial topology of the original high-dimensional structure.

The authors also propose approximate embeddings (circle-based and Voronoi-based) for more efficient visualization, and they validate topological fidelity using persistent homology.

**Strengths:**

The contribution of this paper is to reinterpret the structure of decision trees through a topological lens, showing how the adjacency graph of leaf regions can be visualized in a planar way while (approximately) preserving adjacency.

The paper combines geometry, topology, and model interpretability in a novel way, offering a fresh perspective on a well-studied problem.

The text is clear at a high level, but the mathematical intuition and writing could be improved.

**Weaknesses:**

I have several questions and concerns about the theoretical framing. I should note that I am not an expert in topology, so these are raised mainly as requests for clarification.

While the framing of the discussion is interesting, some of the theoretical claims are confusing. The concept of a "homeomorphic embedding" into $(\mathbb{R}^2\)$ for dimensions $(d\ge 3)$ seems problematic, as such a map cannot actually be a homeomorphism due to the invariance of domain. Instead, the results might describe a combinatorial equivalence between 2-skeletons rather than a full topological equivalence.

Similarly, Lemma 5.1 asserts that any finite graph can be made planar through edge subdivision. This claim appears inconsistent with classical results, as planarity is preserved under subdivision according to Kuratowski (1930). If planarity is necessary, the process may need to alter adjacencies, which contradicts assertions in the paper.

The approximate embeddings are intuitive, but their accuracy is not quantified.

More importantly, beyond toy examples, I don't understand what the main application is for this type of embedding.

Kuratowski, K. Sur le problème des courbes gauches en topologie. Fundamenta Mathematicae, 1930.

**Questions:**

Can the authors show an example of the use case with real data? Illustrating some added value for the user


Could the authors clarify whether the claimed “homeomorphic embedding” is meant in a strict topological sense?

How does the method handle nonplanar adjacency graphs?


How well do the approximate methods preserve adjacency empirically?

---

### Official Review · Reviewer_LAYW · 2025-11-01

**Soundness:** 1
**Presentation:** 2
**Contribution:** 1
**Rating:** 0
**Confidence:** 5

**Summary:**

The paper discusses planar embeddings of decision trees.
The paper claims these exist iff the tree is 2-deep.
It provide mathematical formulaiton of the problem with constructive proof.
Variats of the problem are explored.
This focusses on a visualisation aspect of learning representations, rather than the actual infernece a niche that can have value.
However,  key results are mathematically false, so this cannot be published

**Strengths:**

* refreshing topic

**Weaknesses:**

* Theorem 4.3 is false. One can construct trees of depth >2 that can have a planar representaiton
* Lemma 5.1 directly contradicts the forbiddne subgraph characterization of planar graphs. If this lemma is correct, then this manuscript deserves publicaiton in a much higher impact venue than ICLR.

**Questions:**

Sorry to be blunt but I suggest the authors would check their proofs before submission

---

### Note · Authors · 2025-11-12

I have read and agree with the venue's withdrawal policy on behalf of myself and my co-authors.